# Galactooligosaccharide Mediates NF-κB Pathway to Improve Intestinal Barrier Function and Intestinal Microbiota

**DOI:** 10.3390/molecules28227611

**Published:** 2023-11-15

**Authors:** Menglu Xi, Guo Hao, Qi Yao, Xuchang Duan, Wupeng Ge

**Affiliations:** 1College of Food Science and Engineering, Northwest A&F University, Yangling 712100, China; ximegnlu@sina.cn (M.X.); qiyao@nwafu.edu.cn (Q.Y.); 2Shaanxi Sheep Milk Product Quality Supervision and Inspection Center, Xi’an 710000, China; 15129439429m@sina.cn

**Keywords:** galactooligosaccharides, short-chain fatty acids, gut microbiota, intestinal barrier

## Abstract

The use of antibiotics to treat diarrhea and other diseases early in life can lead to intestinal disorders in infants, which can cause a range of immune-related diseases. Intestinal microbiota diversity is closely related to dietary intake, with many oligosaccharides impacting intestinal microorganism structures and communities. Thus, oligosaccharide type and quantity are important for intestinal microbiota construction. Galactooligosaccharides (GOS) are functional oligosaccharides that can be supplemented with infant formula. Currently, information on GOS and its impact on intestinal microbiota diversity and disorders is lacking. Similarly, GOS is rarely reported within the context of intestinal barrier function. In this study, 16S rRNA sequencing, gas chromatography, and immunohistochemistry were used to investigate the effects of GOS on the intestinal microbiota and barrier pathways in antibiotic-treated mouse models. The results found that GOS promoted *Bifidobacterium* and *Akkermansia* proliferation, increased short-chain fatty acid levels, increased tight junction protein expression (occludin and ZO-1), increased secretory immunoglobulin A (SIgA) and albumin levels, significantly downregulated NF-κB expression, and reduced lipopolysaccharide (LPS), interleukin-IL-1β (IL-1β), and IL-6 levels. Also, a high GOS dose in ampicillin-supplemented animals provided resistance to intestinal damage.

## 1. Introduction

Intestinal damage arises from the destruction of tight junctions at intestinal barriers, permitting lipopolysaccharide (LPS) entry into the blood and causing adverse downstream reactions [1,2]. These include increased mucosal permeability, mucosal inflammation, anemia and malabsorption, protein loss, ileal dysfunction, diarrhea, mucosal ulceration, strictures due to diaphragm disease, and active bleeding and perforation [3].

Approximately 1 × 10^14^ bacterial cells are present in the adult intestine and constitute the gut microbiome [4]. The metabolic capacity of the gut microbiota equals that of the liver; thus, the intestinal microbiota is considered an additional organ [5]. The gut microbiome also maintains host health; many studies have shown that gut microbiome disorders are related to inflammatory bowel disease, obesity, and diabetes [6,7]. Inflammatory bowel disease may lead to intestinal barrier dysfunction; however, ampicillin is also another factor that induces an impaired intestine [8].

Antibiotic abuse increases bacterial resistance, increases infection rates, and culminates in adverse downstream reactions. National statistics have indicated that the number of deaths due to antibiotic abuse is approximately 200,000 each year [9]. The long-term or large-scale use of complex antibiotics diminishes their effects, leads to their accumulation in the body, and causes gastrointestinal microbiota dysbiosis and several associated diseases [10,11]. Long-term antibiotic exposure alters host intestinal bacterial levels and decreases beneficial bacteria quantity and type [12]. Early antibiotic use in infants can also cause intestinal flora disorders and lead to immune and metabolic diseases [13]. Therefore, it is important to identify safe and natural foods that alleviate and possibly prevent gut microbiome disorders.

Prebiotics are selectively fermented components that elicit specific changes in the composition and/or activity of gastrointestinal microflora, thereby imparting host health benefits. The beneficial bacteria mostly targeted by prebiotics are almost exclusively *Bifidobacterium* and *Lactobacillus* [14]. Galactooligosaccharides (GOS) are recognized as food additives; they are indigestible, so they may be used by *Bifidobacterium* to generate and secrete short-chain fatty acids (SCFAs) [15,16]. Studies have found that adding prebiotics early in life can help infants establish a healthy gut microbiome [17]. At the same time, the direct or combined use of prebiotics in the treatment of infant diarrhea is more effective than the use of antibiotics alone and can effectively prevent intestinal disorders caused by antibiotics, etc., which also suggests that the use of prebiotics to correct intestinal microecological disorders may become an important measure to improve antibiotic treatment [18]. However, it is currently unclear whether GOS alleviates intestinal barrier damage by promoting probiotic proliferation, and similarly, few studies have investigated the effect of GOS on preventing and recovering microbiota after ampicillin-induced microbiota disruption.

Here, we studied the effects of GOS on the intestinal microbiota of weanling mice. We assessed whether GOS improved the gut microbiota of ampicillin-affected mice, induced effects in early postnatal life, and explored mechanisms underpinning GOS-mediated regulation of ampicillin exposure.

## 2. Results

### 2.1. GOS Alters the Composition of the Intestinal Microbiota in Weaned Mice after Antibiotic Treatment

To explore the effects of ampicillin and GOS on the intestinal microbiota, we performed 16S rRNA sequencing. Rarefaction and rank curves are commonly used to describe sample diversity within a group. The rarefaction curve (Figure 1a,b) directly reflected the sufficiency of the amount of sequencing data and indirectly reflected species richness in samples. As the curve flattens, the increase in the amount of sequencing data gradually decreases, and the inclusion of more data produces only a small number of new species (OTUs). From species annotation analyses, the top 10 species with the highest abundance in each sample or group at each classification level (phylum and genus) were selected, and a column of the relative species abundance was generated to view each sample at different taxonomic levels. Then, relatively abundant species and their proportions were identified [19]. At the phylum level (Figure 1c), the intestinal microbiota mainly consisted of *Firmicutes*, *Bacteroidetes*, *Proteobacteria*, and *Actinobacteria*. Genus-level differences were identified after 8 and 12 weeks of GOS intake (Figure 1d). At Week 8, the relative abundance of *Firmicutes* in the intestinal microbiota of all antibiotic-treated groups relative to non-antibiotic groups decreased, and the relative abundance of *Actinobacteria* increased. Compared with the control group, the relative abundance of *Firmicutes* in the FPC, FPL, FPM, and FPH groups after penicillin treatment decreased by 20.83%, 45.41%, 42.36%, and 36.11%, respectively. The relative abundance of *Bacteroidetes* in the FPC, FPL, FPM, and FPH groups increased by 27.22%, 108.11%, 134.05%, and 37.83%, respectively. The relative abundance of *Actinobacteria* in the FPC, FPL, FPM and FPH groups increased by 175%, 274.28%, 201.04%, and 403.93%, respectively. The relative abundance of *Proteobacteria* in the FPC, FPL and FPM groups decreased by 66.67%, 22.22%, and 49.45%, respectively, while the relative abundance in the FPH group increased by 113.89%. In addition, after penicillin treatment, the relative abundance of *Firmicutes* in the GOS group showed a downward trend regardless of whether the low, medium, or high dose was used. A possible explanation for this was that penicillin inhibits chitosan synthesis in Gram-positive bacteria cell walls, thus inhibiting growth. *Firmicutes* are Gram-positive bacteria, so their numbers were reduced. The dominant bacterial composition of FMC and FMM groups treated with mixed antibiotics was similar at the phylum level and mainly consisted of *Firmicutes*, *Bacteroidetes*, and *Proteobacteria*.

At Week 12, the relative abundance of *Firmicutes* in the intestinal microbiota of mice without penicillin decreased, and *Actinobacteria* increased in the medium-dose group and high-dose group after penicillin treatment. SPC and SPL groups were relative to SC and SL groups. The relative abundances of *Actinobacteria* and *Verrucomicrobia* in SPM and SPH groups increased significantly from 6.23% to 16.99% and 0.55% to 6.29%, and 6.31% to 26.45% and 0.81% to 5.68%, respectively; the relative abundances of *Dubosiella* and *Faecalibaculum* in the FC group were 12.20% and 18.62%; FPC group, 1.56% and 0.78%; FL group, 32.05% and 31.77%; FPL group, 2.52% and 2.21%; FM group, 24.16% and 7.61%; FPM group, 2.29% and 0.90%; FH group, 14.72% and 8.63%; and FPH group, 0.12% and 0.10%, respectively. The relative abundance of *Dubosiella* and *Faecalibaculum* in the normal group decreased significantly, while in the penicillin group, their relative abundance decreased almost to zero, suggesting that penicillin exerted particularly significant inhibitory effects on these strains. The relative abundance of *Akkermansia* in the FL group decreased from 8.68% to 3.48%, but its relative abundance in the FL group increased from 1.93% to 4.77%. The relative abundance of *Bifidobacterium* in the same FC group was 1.82%; the abundance of *Bifidobacterium* in the FPC group was 3.50%. The relative abundances of *Klebsiella*, *Enterococcus*, and *Bacteroides* in the FMC and FMM groups were 35.52%, 22.16% and 15.28%; 30.67%, 28.25% and 9.05%, suggesting mixed antibiotic treatments effectively killed large numbers of beneficial bacteria in the intestine. At the genus level, the dominant bacteria in the mixed antibiotic control group were *Clostridioides*, *Enterococcus*, and *Blautia*. The dominant bacteria in the mixed antibiotic medium GOS dose group were *Klebsiella*, *Clostridioides*, and *Enterococcus*, suggesting that mixed antibiotic treatments were effective. *Akkermansia* and *Enterococcus* were the dominant bacteria in the medium GOS dose group exposed to penicillin, and *Akkermansia*, *Lactobacillus*, *Bifidobacterium*, and *Faecalitalea* were the dominant bacteria in the high GOS dose group exposed to penicillin. *Faecalibaculum* abundance in the control group decreased, whereas medium and high GOS doses promoted *Bifidobacterium* and *Lactobacillus* abundance to different degrees. In the mixed antibiotic control group, *Enterococcus* and *Klebsiella* abundance increased significantly. When compared with the mixed antibiotic control group, a medium GOS dose was not effective to the abundance of the mixed antibiotics *Enterococcus* and *Klebsiella*.

Using linear discriminant analysis (LDA), we analyzed species with significant abundance differences in different groups. At 8 weeks (Figure 1e,f), the different species between the medium-dose GOS group and the other groups were *Prevotellaceae,* and the difference in the high GOS group was *Bifidobacterium* compared with other groups. In similar conditions, at Week 12, the different species between the medium-dose GOS group and the other groups were *Akkermansiaceae,* and the difference in the high GOS group was *Bifidobacterium* compared with other groups, which showed that GOS can promote the proliferation of *Bifidobacterium*.

### 2.2. GOS Alters SCFA Production in the Feces of Weaned Mice after Antibiotic Treatment

Metabolite expression is closely associated with intestinal microbial function [19]. As we showed that intestinal microbiota levels changed in our model, we hypothesized that microbial metabolite production also changed; therefore, we measured acetic, propionic, and butyric acid levels. Cecal SCFA levels were monitored throughout the study as indicators of metabolite production (Figure 2a–c).

Upon measuring acetic acid levels, levels in the ampicillin-exposed control group were lower than those in the normal control group; however, the reduction was not significant. The medium-dose GOS group showed significantly increased acetic acid levels (*p* < 0.05) and suggested this state promoted acetogenic bacterial growth and colonization. In addition, acetic acid levels decreased significantly after treatment with mixed antibiotics (*p* < 0.05). This suggested that most of the acetic acid-producing bacteria in the intestine were killed or reduced. In combination with 16S rRNA sequencing results, these results also verify our hypothesis [19].

Trend changes in propionic acid levels were consistent with acetic acid levels. However, the medium GOS dose increased propionic acid levels and suggested an increased proliferation of propionic acid-producing bacteria [19]. At 12 weeks, propionic acid levels in the ampicillin-exposed group were significantly lower than those in the normal control group (*p* < 0.05). In the mixed antibiotic group, sharp decreases in propionic and butyric acid levels were observed. A high GOS dose exerted certain protective effects under the exposition of ampicillin and significantly increased propionic and butyric acid levels (*p* < 0.05). Bacterial 16S rRNA sequencing results showed that medium-dose GOS significantly increased *Akkermansia* growth, and a high GOS dose promoted *Bifidobacterium* abundance; thus, acetic and propionic acids were the primary metabolites of *Akkermansia* and *Bifidobacterium*, even under the exposition of ampicillin.

### 2.3. GOS Improves the Intestinal Barrier and Reduces Colon Tissue Inflammation after Antibiotic Treatment

Ampicillin damages the intestinal microbiota, resulting in microbiota disorders and the destruction of intestinal barrier integrity. Probiotics help maintain intestinal barrier function, while GOS promotes probiotic colonization. As GOS promoted the proliferation of beneficial bacteria, we investigated whether ampicillin-mediated colon damage was reduced after GOS intake. Hence, intestinal morphology was used to assess whether beneficial bacteria were associated with gut morphology (Figure 3). When compared with control group C, control group PC, exposed to penicillin, displayed epithelial cell abscission and necrosis, accompanied by low congestion, neutrophil infiltration, and low numbers of goblet cells. In the MC of the control group exposed to mixed antibiotics, goblet cells were significantly decreased and accompanied by some congestion, glandular epithelial cells fell off, lamina propria were seriously atrophied, a large number of crypts disappeared, and a muscle layer fell off. During penicillin exposure, the score in the low-dose GOS treatment group was higher than that in the PC group, but with a small amount of hyperemia. In the PM group and the pH group, inflammatory cell infiltration decreased, and epithelial cell villi arrangement was orderly. Also, there were more goblet cells in the pH group. In the high-dose GOS group, intestinal immune functions were stronger. Thus, medium and high GOS doses increased beneficial bacteria proliferation to protect intestinal morphology, but the doses had no significant effects on intestinal injury after mixed antibiotic treatment.

As we observed morphology damage in the intestine, serum inflammatory factors were assayed by ELISA to assess damage levels. When mice were exposed to ampicillin, medium and high GOS doses resulted in increased serum LPS and TNF-α levels, to varying degrees (Figure 4a–c). Medium and high GOS doses also significantly reduced serum IL-17 levels in mice (*p* < 0.01), whereas low and medium doses reduced TNF-α serum levels (*p* < 0.01), thereby promoting anti-inflammatory effects. Also, medium and high GOS doses significantly reduced serum LPS levels (*p* < 0.01). Thus, a specific GOS intake reduced inflammation levels, suggesting protective and alleviation effects on intestinal barrier function.

We also measured tight junction protein expression in the intestine. Using immunohistochemistry (Figure 5), the ampicillin-exposed group had a significantly reduced occludin expression in the intestine. However, the GOS-intervention group had an increased occludin expression, suggesting GOS increased tight junction protein expression at these sites. Therefore, GOS intake affected tight junction protein expression in the colon of penicillin-treated mice. Occludin and ZO-1 expression were promoted, and the promotion effect of pH in the high-dose group was the most significant, indicating that GOS can reduce the colonic injury caused by penicillin and repair the intestinal barrier. Also, tight junction protein expression in the MM group was similar to that of the M group, suggesting GOS did not repair intestinal barriers when large numbers of intestinal microbiota were removed.

The contents of SIgA and albumin were significantly decreased in the PC group, and a certain dose of GOS could promote the increase in SIgA and albumin levels (Figure 6a,b). We speculated that antibiotics may have destroyed some of the intestinal microbiota and removed key intestinal bacteria groups responsible for the generation of substances that altered SIgA and albumin levels.

The NF-κB pathway is a key inflammatory release pathway. Therefore, to explore GOS-mediated regulatory inflammatory mechanisms, we performed immunohistochemical analyses in colonic tissue to characterize NF-κB pathway changes (Figure 7). Penicillin or mixed antibiotics induced NF-κB overexpression (*p* < 0.05). PH significantly downregulated NF-κB expression (*p* < 0.05), suggesting that GOS may reduce penicillin-induced inflammatory factors by down-regulating the expression of NF-κB. We observed no significant differences between the MC and MM groups, which suggested GOS could not improve inflammatory pathway functions when intestinal microbiota were largely cleared.

## 3. Discussion

We showed that GOS promoted *Bifidobacterium*, *Lactobacillus*, and *Akkermansia* proliferation. In previous studies, GOS promoted the proliferation of beneficial bacteria such as *Bifidobacterium* and *Lactobacillus* in the intestine, consistent with Wu [20], who also showed that GOS promoted *Lactobacillus*. Krumbeck et al. also showed that GOS promoted *Bifidobacterium* and *Lactobacillus* proliferation in the intestines of obese adults and improved intestinal barrier function [21].

After ampicillin treatment, the relative abundance of *Dubosiella* and *Faecalibaculum* in the normal group was significantly decreased, while abundance in the ampicillin-exposed group decreased to zero, suggesting that ampicillin had a significant inhibitory impact on these strains. *Klebsiella*, *Clostridioides*, and *Enterococcus* were the dominant bacteria in the mixed antibiotic group. Martin et al. reported that *Klebsiella* more easily colonized the intestinal tract under antibiotic conditions, suggesting mixed antibiotics were highly effective in killing most intestinal microorganisms and reducing the relative abundance of beneficial bacteria [22]. When microbiota were ablated by mixed antibiotics, bacterial infections were more likely to occur, and infection resistance was compromised. Therefore, intestinal microbiota are an important barrier to protecting the host from bacterial infections; thus, antibiotic use should be minimized. At 8 and 12 weeks, the relative abundance of *Bifidobacterium* was increased in the high-dose treatment group under the exposition of ampicillin. Soldi et al. reported that antibiotic use reduced the relative abundance of *Bifidobacterium* in the intestinal microbiota of children, and the level of *Bifidobacterium* in probiotics and patients treated with antibiotics was significantly higher than that in children receiving placebo control, consistent with our research [23].

From H&E analyses of PM and PH groups, inflammatory cell infiltration was reduced, suggesting that medium and high GOS doses protected the intestinal morphology of mice with intestinal disorders, and effectively alleviated intestinal barrier damage, permitting improved intestinal immune function.

SCFAs, which are produced by beneficial bacteria, not only lower intestinal pH and inhibit harmful bacteria growth, but also correlate with the immune system. When we combined 16S rRNA sequencing with SCFA analysis, medium- and high-dose GOS promoted the proliferation of acetic acid, propionic acid, and butyric acid-producing bacteria in mice intestines, thereby increasing the levels at these sites. Intestinal cell proliferation is closely related to intestinal absorption function. Yang et al. found that tributyrin can restore dysregulation of intestinal flora by releasing butyric acid, inhibit inflammation, and improve antibiotic-induced intestinal damage and barrier function, consistent with our results [24].

Guilloteau et al. supplemented calf diets with flavomycin or sodium butyrate to assess villus length in intestinal segments; sodium butyrate significantly increased the length of the duodenum and jejunum villi when fed to animals compared with the results obtained from flavomycin-fed animals; thus, exogenous sodium butyrate increased intestinal villi length [25]. The reason is that sodium butyrate increases the absorption area of nutrients and the intestine, while sodium butyrate is the function of butyric acid in the intestine, which is closer to our results.

Impaired intestinal barrier function leads to impaired functional output; therefore, we measured serum inflammatory factors in mice and showed that TNF-α levels in low- and medium-GOS treatment groups were significantly lower than in the control group (*p* < 0.01). Also, LPS levels in the high-dose GOS treatment group were significantly lower than in the control group. Previous studies reported that *Prevotella*, *Parabacteroides*, *Clostridium*, and *Adlercreutzia* were considered anti-inflammatory symbioses [26], and in the 16SrRNA results, it was found that the common dominant bacteria of the SPL group and the SPM group all have *Parabacteroides*, and an advantage was found in the SPM group. Bacteria such as *Clostridium* and *Parabacteroides* may be a factor in reducing inflammation.

We also determined SIgA and albumin levels in mouse feces to assess immune factor levels in the intestines. We previously observed that both ampicillin sodium and mixed antibiotics destroyed diversity and the relative abundance of intestinal microbiota; therefore, we speculated that the part of the intestinal microbiota that is cleared may produce important substances. After being cleared, it may cause changes in the levels of intestinal immune factors SIgA and albumin. Mantis et al. indicated that SIgA was the first line of defense in protecting internal environments from internal toxins and pathogens and directly eliminated bacterial virulence [27]. When a particular intestinal microbiota is eliminated or destroyed, the body’s ability to produce SIgA and albumin is reduced, thereby affecting immune function at the intestinal barrier. We observed that SIgA and albumin levels in the ampicillin-treated group were lower than the normal group, consistent with our experimental hypothesis. The clearance of some intestinal microbiota decreased SIgA and albumin levels. However, we observed no significant differences in SIgA and albumin levels between the MC and MM groups, and albumin levels were lower than those in the control group, which also indicated that the specific part of the intestinal microbiota was removed. The microbiota is the central link between GOS and albumin. SIgA levels in the MC and MM groups were higher than those in the control group. Calguneri et al. reported that the relative abundance of *Klebsiella* was found in the intestinal microbiota of patients with ankylosing spondylitis, as well as in the salivary SIgA and serum IgG and IgA [28]. The average value of C3 and C4 increased significantly. In their study, Davis et al. reported that the fimbriae of *Klebsiella* stimulated the specific binding of B cells to SIgA [29].

In summary, we characterized LPS, TNF-α, and IL-17 levels using histopathological and morphological observations of H&E sections, ELISA, and LPS levels to preliminarily investigate the alleviating effects of GOS on the intestinal barrier of mice with an intestinal disorder. In PM and PH group H&E sections, inflammatory cell infiltration was decreased, suggesting that medium and high GOS doses protected intestinal morphology in mice with intestinal disorders, thereby effectively alleviating intestinal barrier damage and offering better play to intestinal immune function. When combined with 16S rRNA sequencing and SCFA data, medium and high GOS doses promoted and increased acetic acid, propionic acid, and butyric acid-producing bacterial levels in the intestinal tract of mice. We also analyzed serum inflammatory factors in mice and observed TNF-α in low- and medium-dose GOS treatment groups. LPS levels were significantly lower than those in the control group, and the content of LPS treated with medium and high doses of GOS was significantly lower than that in the control group. From 16S rRNA data, the most common dominant bacteria in the SPL and SPM groups were *Parabactoids*, and the most dominant bacteria in the SPM group were *Clostridium*. *Parabactoids* may be a factor in reducing the inflammatory effect. LPS is a cell wall component of Gram-negative bacteria; it enters the blood circulation when the tight connections in the intestinal barrier are damaged, thereby causing inflammatory reactions. Penicillin inhibits chitosan synthesis in bacterial cell walls and prevents the proliferation of most Gram-positive bacteria, leading to an imbalance between Gram-positive and Gram- negative bacteria. GOS mainly promotes the proliferation of the beneficial bacteria *Bifidobacterium,* which are Gram-positive bacteria. Combined with the analysis of 16S rRNA results, with the increase in the GOS dose, the proportion of remaining Gram-positive bacteria in the mouse intestinal microbiota is relatively high, which improves the relative abundance of Gram-negative bacteria in low- and medium-dose GOS treatment groups, which may cause the LPS level in the SPL and SPM groups to become higher than that in the SPH group. The above results show that GOS significantly improved intestinal morphology, increased SCFA production, increased tight junction protein expression, and reduced intestinal inflammation levels. A high GOS dose in the ampicillin group provided some resistance to intestinal damage.

## 4. Materials and Methods

### 4.1. Animals and Experimental Design

Three-week-old male C57BL/6J mice were acquired from Xi’an Jiaotong University Health Science Center (Xi’an, China) and housed in filter-top cages [19]. Forty mice were randomly divided into five groups (*n* = 8/group) and received (i) a normal AIN-93G diet and water (C); (ii) a normal AIN-93G diet and water containing ampicillin (1 g/L, PC), Sigma-Aldrich Corp. (St. Louis, MO, USA); (iii) a low-dose GOS AIN-93G diet (0.5% *w*/*w*) and water containing ampicillin (1 g/L, PL), Sigma-Aldrich Corp. (St. Louis, MO, USA); (iv) a medium-dose GOS AIN-93G diet (2% *w*/*w*) and water containing ampicillin (1 g/L, PM), Sigma-Aldrich Corp. (St. Louis, MO, USA); and (v) a high-dose GOS AIN-93G diet (5% *w*/*w*) and water containing ampicillin (1 g/L, PH), Sigma-Aldrich Corp. (St. Louis, MO, USA).

Also, other mice were randomly divided into two groups (*n* = 10/group) and received (i) a regular AIN-93G diet and water containing streptomycin (1 g/L), ampicillin (1 g/L), and gentamicin (1 g/L) (Sigma-Aldrich Corp. (St. Louis, MO, USA)) to clear intestinal bacteria (MC); and (ii) a medium-dose GOS AIN-93G diet (2% *w*/*w*) and water containing streptomycin (1 g/L), ampicillin (1 g/L), and gentamicin (1 g/L)(MM) [19] (Table 1).

At the end of Weeks 8 (F) and 12 (S), feces were collected, frozen in liquid nitrogen, and placed at −80 °C. At the end of Week 12, mice were humanely sacrificed (09:00 h) to collect cecum and colon samples.

Animal experiments were conducted in accordance with the National Institutes of Health Guidelines for the Care and Use of Laboratory Animals (NIH Publication no. 80-23) and approved by the Xi’an Jiaotong University and Northwest A&F University Animal Care and Use Committee (Yangling, Shaanxi, China). The approval number is No. XJTULAC2014-1035, and a member of the ethics committee that approved this study is Teng Chen. Experimental mice were maintained in controlled environmental conditions: 25 °C ± 1 °C ina 55% ± 15% relative humidity, and a 12/12 h light–dark cycle [19].

### 4.2. SCFA Quantification in Feces

SCFA (acetate, propionate, and butyrate) levels were quantified using a Shimadzu GC-2014C gas chromatograph (Shimadzu Corporation, Kyoto, Japan) equipped with a DB-FFAP capillary column (30 m × 0.25 m × 0.25 mm) (Agilent Technologies, Santa Clara, CA, USA) and a flame ionization detector [30]. Approximately 400 mg of feces was homogenized in 1.6 mL of distilled water, with 0.4 mL of 50% H_2_SO_4_ (*w*/*w*) and 2 mL of diethyl ether added. Samples were incubated at 4 °C for 30 min and centrifuged at 10,000× *g* for 5 min. The organic phase was collected and analyzed using gas chromatography under the following conditions: an initial temperature of 50 °C was maintained for 1 min and raised to 120 °C at 15 °C/min, followed by an increase to 170 °C at 5 °C/min, and then increased to 240 °C at 15 °C/min. This was maintained for 3 min. Injector and detector temperatures were 250 °C and 270 °C, respectively [19].

### 4.3. 16S rRNA Sequence Analysis

DNA was extracted from feces using a Qiagen QIAamp DNA Stool Mini kit (Qiagen, Germany). The library was sequenced on an Ion S5TMXL platform, and raw reads were quality filtered under specific filtering conditions to generate high-quality clean reads using Cutadapt. In total, 7,699,625 clean tags were subjected to the following analysis. Reads were compared with a reference database using the UCHIME algorithm to detect chimeric sequences which were removed. Sequence analysis was performed using Uparse (Uparse v7.0.1001) and SILVA reference databases. The primers used to amplify the 16S rRNA gene (V3-V4) were 5′-ACTCCTACGGGAGGCAGCA-3′ (338F) and 5′-GGACTACHVGGGTWTCTAAT-3′(806R). Sequences with ≥97% similarity were assigned to the same operational taxonomic unit (OTU). Beta diversity analysis was used to evaluate sample differences with respect to species complexity. Beta diversity was assessed by both weighted and unweighted UniFrac analysis, calculated by QIIME software (Version 1.7.0) [19].

### 4.4. RNA Extraction and RT-qPCR Analysis

Following sacrifice, the distal ileum and colon were quickly dissected and stored in an RNA buffer at −80 °C. Following tissue homogenization, total RNA was extracted using TRIzol (Bioteke, Beijng, China). Then, 1 μg RNA was converted to cDNA using the SuperscriptIII First-Strand Synthesis Supermix kit (Bioteke, Beijng, China). Diluted cDNA was used for qPCR using the PowerUp SYBR Green Master Mix kit (Bioteke). qRT-PCR was performed using fast mode Dual-Lock DNA polymerase at 95 °C for 10 min, followed by 40 cycles of 95 °C for 15 s, 60 °C for 30 s, and 72 °C for 15 s. Real-time PCR was performed using a CFX96 Real-Time PCR Detection System (Bio-Rad, Hercules, CA, USA). Data were normalized to the endogenous control, GAPDH, with relative quantification performed using the 2-ΔΔCt method [19]. The primers used were described by Leclercq [8]. All experiments were performed in triplicate.

### 4.5. Histomorphology and Immunohistochemistry

Approximately 1 cm of the proximal colon was fixed overnight in 4% paraformaldehyde at 4 °C. Colon tissues were embedded in paraffin and sectioned into 5 μm slices and stained in hematoxylin and eosin (H&E) to evaluate colonic architecture, crypt loss, mucosal damage, and lymphocyte infiltration.

To investigate tight junction proteins, immunohistochemical staining was performed [30]. After deparaffinization, sections underwent heat-mediated antigen retrieval in a citrate buffer (pH 6.0) at 95–100 °C before incubation overnight with polyclonal anti-occludin and monoclonal anti-ZO-1 antibodies at 4 °C. Then, tissues were probed with horseradish peroxidase-conjugated secondary antibodies for 1 h at room temperature. Peroxidase activity was visualized using 3,3-diaminobenzidine chromogen. All sections were examined under a Motic BA400 microscope (Motic Co. Ltd., Guiyang, China) and mean density values assessed using Image J (v1.8.0). Tight junction protein levels and inflammatory pathway protein levels were blindly assessed [19].

### 4.6. Enzyme-Linked Immunosorbent Assay (ELISA)

Serum LPS, tumor necrosis factor-α (TNF-α), and fecal albumin and secretory immunoglobulin A (SIgA) levels were measured using ELISA kits in accordance with manufacturer’s instructions (Bio-Rad, Shanghai, China). ELISA detection ranges for LPS, TNF-α, SIgA, and albumin were 2–32 EU/L, 20–640 pg/mL, 1–32 µg/mL, and 2–240 ng/mL, respectively.

### 4.7. Statistical Analysis

Data were presented as the mean ± standard deviation (SD). Prism 6.0, GraphPad was used for all analyses. Differences at *p* < 0.05 were considered statistically significant.

## Figures and Tables

**Figure 1 molecules-28-07611-f001:**
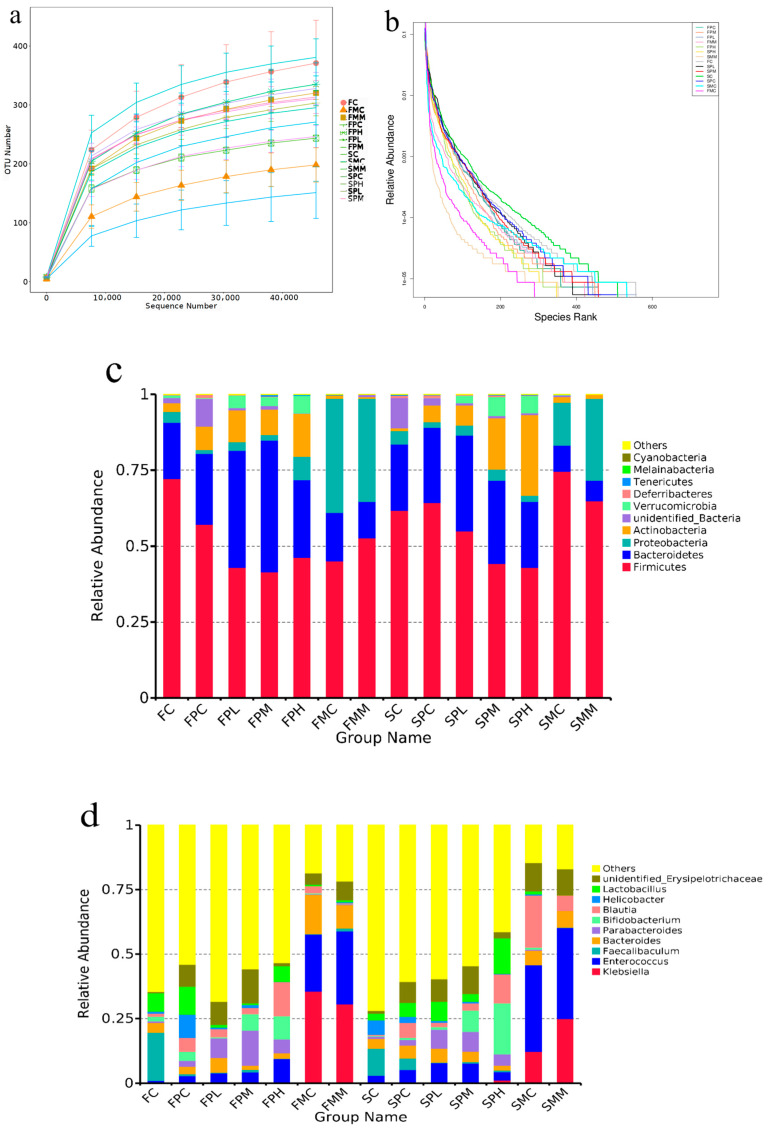
Intestinal microbiota composition after GOS supplementation with ampicillin exposition. (**a**) Rarefaction curve; (**b**) Rank curves; (**c**) Relative contribution of the top 10 phyla in each group; (**d**) Relative contribution of the top 10 genera in each group; (**e**,**f**) Taxonomic cladogram from LEfSe at 8 and 12 weeks, (*n* = 4). Biomarker taxa are highlighted with colored circles and shaded areas. Circle diameters reflect taxa abundance in the community. A cut-off value of ≥4.0, used for linear discriminant analysis (LDA), is shown. F 8 weeks, S 12 weeks.

**Figure 2 molecules-28-07611-f002:**
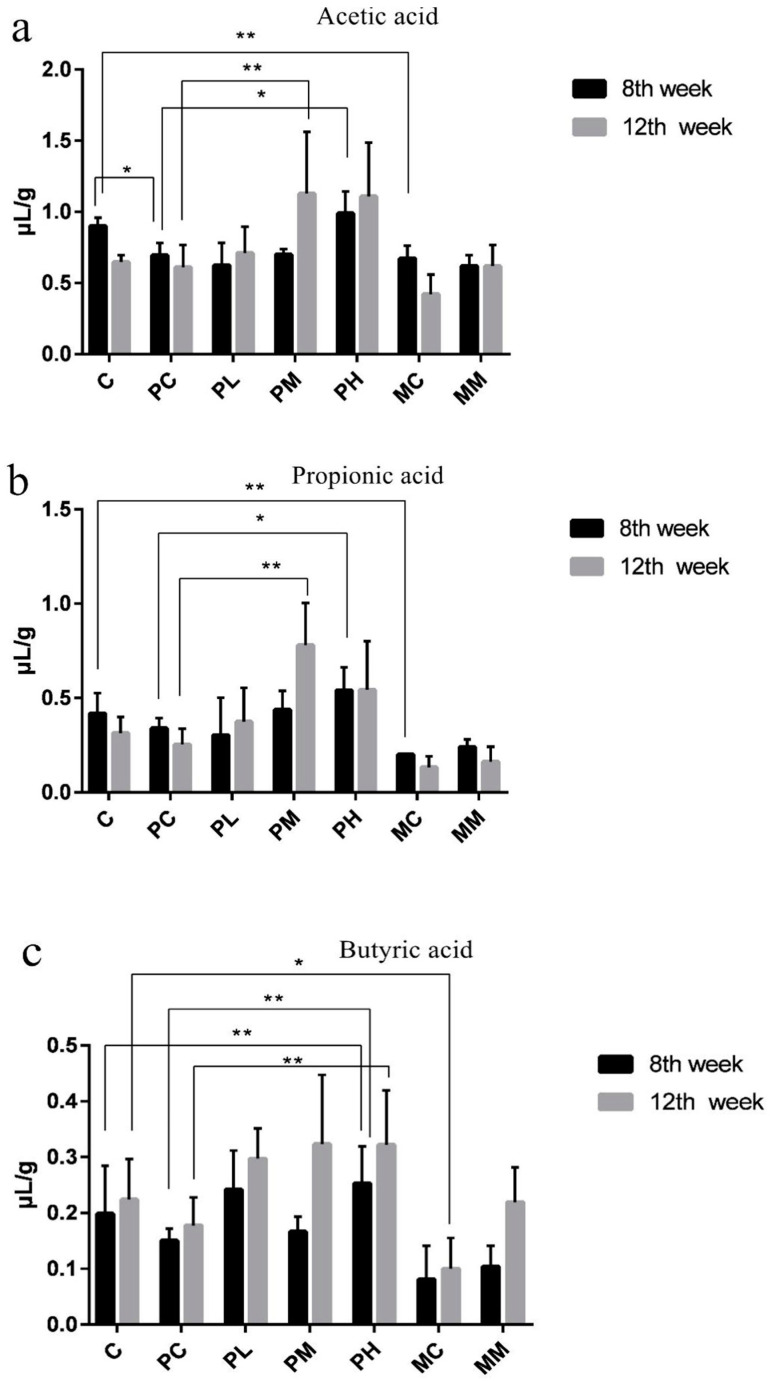
Short-chain fatty acid levels in feces. Acetic (**a**), propionic (**b**), and butyric acid (**c**) levels in cecal contents were measured using gas chromatography at 8 weeks (black) and 12 weeks (light gray) after study commencement (*n* = 8). Data (mean ± standard deviation (SD)) were analyzed using one-way ANOVA (* *p* < 0.05; ** *p* < 0.01). Data are represented by three biological replicates. C is a normal AIN-93G diet and water; PC is a normal AIN-93G diet and water containing ampicillin; PL is a low-dose GOS AIN-93G diet (0.5% *w*/*w*) and water containing ampicillin; PM is a medium-dose GOS AIN-93G diet (2% *w*/*w*) and water containing ampicillin; PH is a high-dose GOS AIN-93G diet (5% *w*/*w*) and water containing ampicillin; MC is a regular AIN-93G diet and water containing streptomycin (1 g/L), ampicillin (1 g/L), and gentamicin (1 g/L); MM is a medium-dose GOS AIN-93G diet (2% *w*/*w*) and water containing streptomycin (1 g/L), ampicillin (1 g/L), gentamicin (1 g/L).

**Figure 3 molecules-28-07611-f003:**
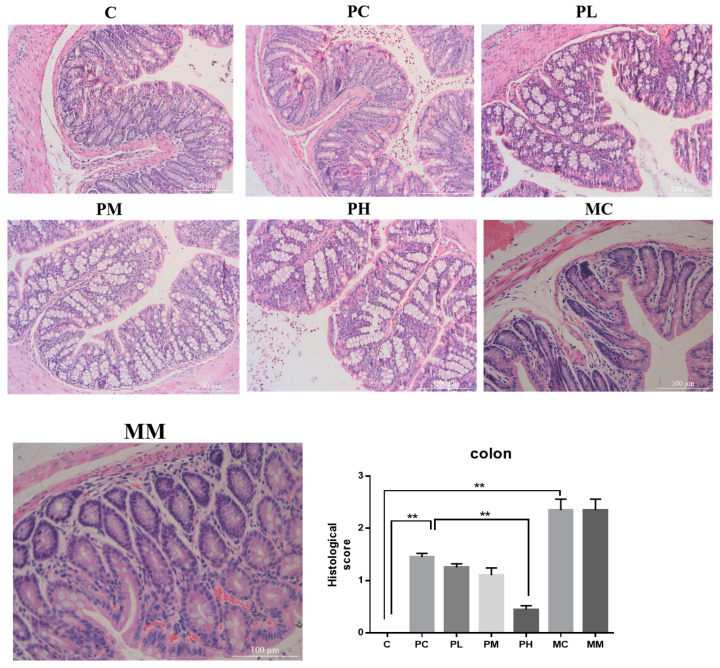
Representative images of hematoxylin and eosin staining (magnification 200×; scale bar = 200 μm); the gray bar chart presents the histological scores of the colon sections (** *p* < 0.01). C is a normal AIN-93G diet and water; PC is a normal AIN-93G diet and water containing ampicillin; PL is a low-dose GOS AIN-93G diet (0.5% *w*/*w*) and water containing ampicillin; PM is a medium-dose GOS AIN-93G diet (2% *w*/*w*) and water containing ampicillin; PH is a high-dose GOS AIN-93G diet (5% *w*/*w*) and water containing ampicillin; MC is a regular AIN-93G diet and water containing streptomycin (1 g/L), ampicillin (1 g/L), and gentamicin (1 g/L); MM is a medium-dose GOS AIN-93G diet (2% *w*/*w*) and water containing streptomycin (1 g/L), ampicillin (1 g/L), gentamicin (1 g/L).

**Figure 4 molecules-28-07611-f004:**
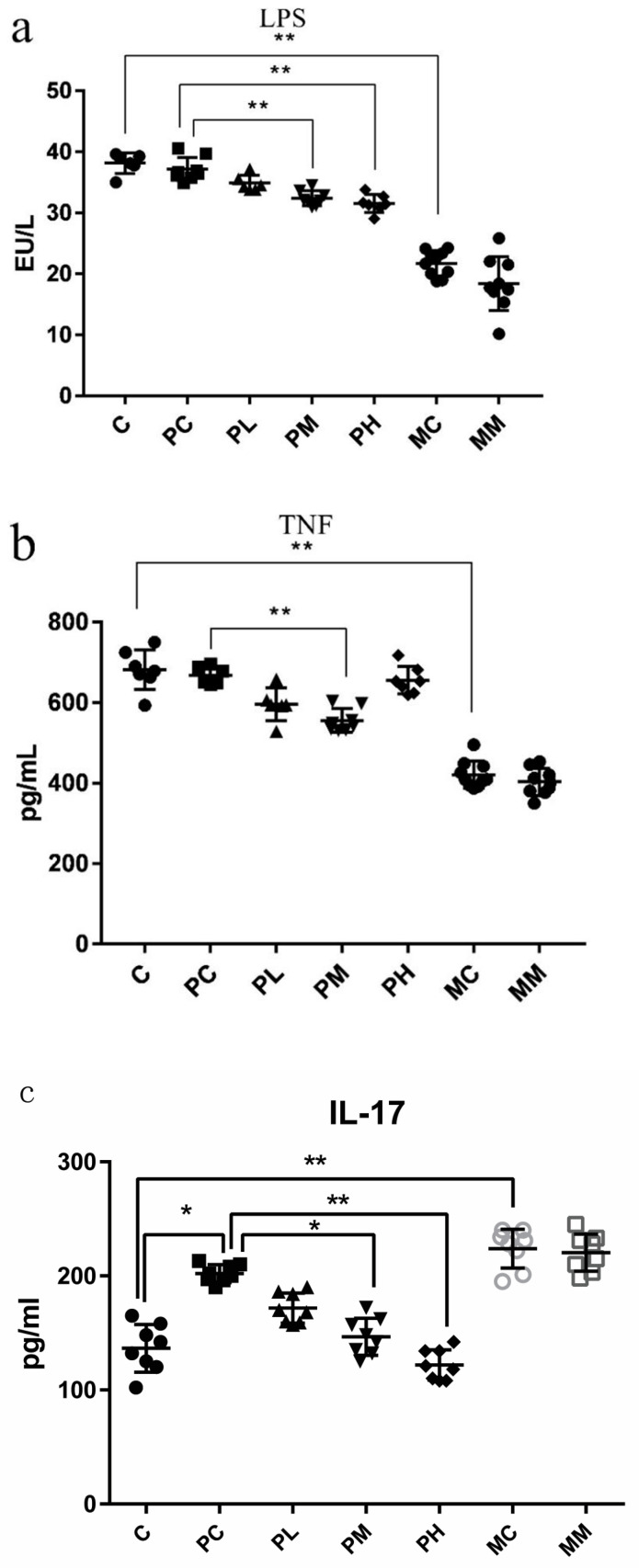
Assessing inflammation, immunocompetence, and intestinal permeability biomarkers using enzyme-linked immunosorbent assay. In (**a**–**c**), the concentrations of LPS, TNF-α, and IL-17 in mouse serum are represented, respectively (*n* = 8). Data (mean ± standard deviation (SD)) were analyzed using one-way ANOVA (* *p* < 0.05; ** *p* < 0.01).

**Figure 5 molecules-28-07611-f005:**
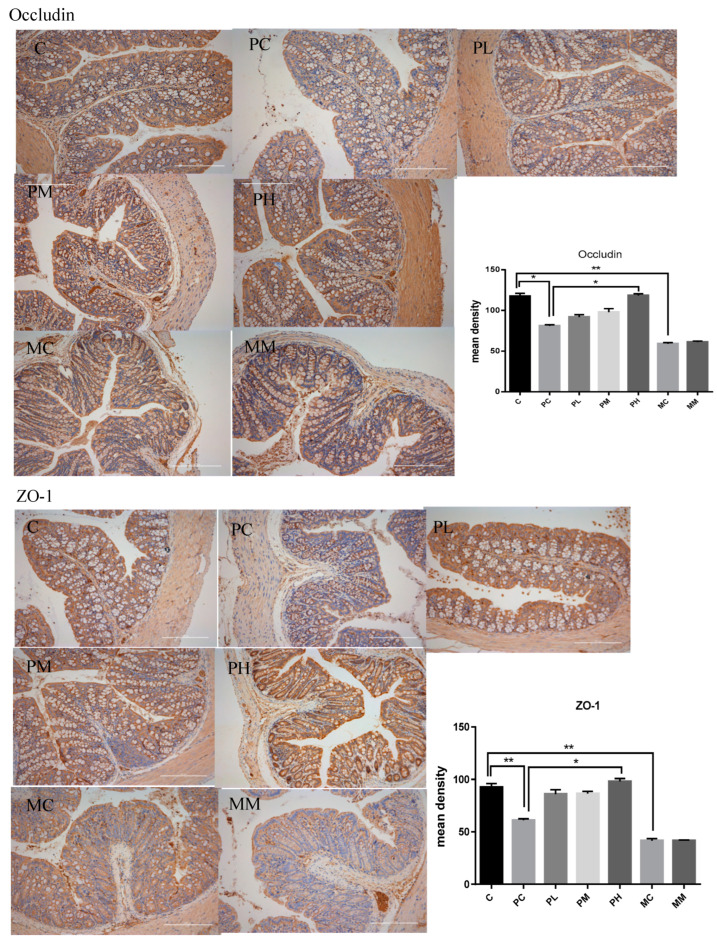
Immunohistochemistry of occludin and ZO-1 in the colon. The bottom panel presents the mean density of immunohistochemistry. Data (mean ± SD) were analyzed with one-way ANOVA (* *p* < 0.05; ** *p* < 0.01).

**Figure 6 molecules-28-07611-f006:**
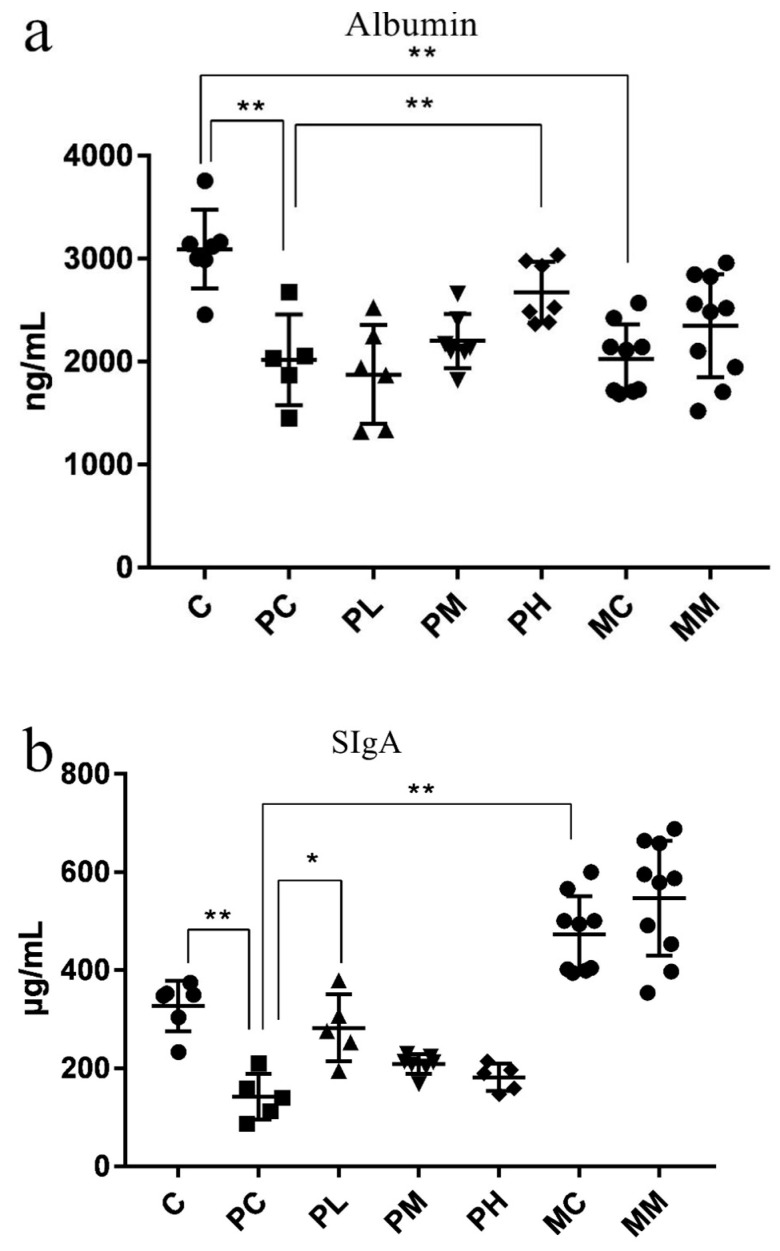
(**a**,**b**) Evaluating intestinal barrier immunity and tight junctions by measuring SIgA and fecal albumin levels (*n* = 8). Data (mean ± standard deviation (SD)) were analyzed using one-way ANOVA (* *p* < 0.05; ** *p* < 0.01).

**Figure 7 molecules-28-07611-f007:**
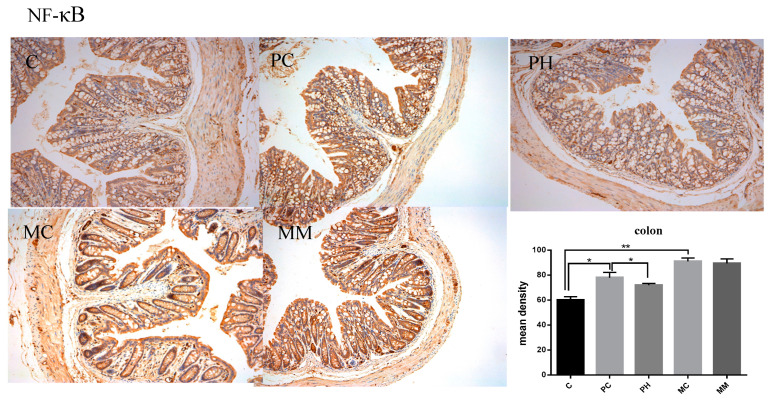
Immunohistochemistry of NF-κB in the colon. The bottom panel presents the mean density of immunohistochemistry. Data (mean ± SD) were analyzed with one-way ANOVA (* *p* < 0.05; ** *p* < 0.01).

**Table 1 molecules-28-07611-t001:** Grouping of test animals.

Group	8 Weeks	12 Weeks	Diet	Water
C	FC	SC	normal AIN-93G	normal water
PC	FPC	SPC	normal AIN-93G	water containing ampicillin (1 g/L)
PL	FPL	SPL	low-dose GOS AIN-93G diet (0.5% *w*/*w*)	water containing ampicillin (1 g/L)
PM	FPM	SPM	medium-dose GOS AIN-93G diet (2% *w*/*w*)	water containing ampicillin (1 g/L)
PH	FPH	SPH	high-dose GOS AIN-93G diet (5% *w*/*w*)	water containing ampicillin (1 g/L)
MC	FMC	SMC	normal AIN-93G	water containing streptomycin (1 g/L), ampicillin (1 g/L), and gentamicin (1 g/L)
MM	FMM	SMM	medium-dose GOS AIN-93G diet (2% *w*/*w*)	water containing streptomycin (1 g/L), ampicillin (1 g/L), and gentamicin (1 g/L)

## Data Availability

The gut microbiome dataset was deposited into the NCBI Sequence Read Archive under accession number SUB5958837.

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
