# Peer review of "Galactooligosaccharide Mediates NF-κB Pathway to Improve Intestinal Barrier Function and Intestinal Microbiota"

_molecules, 2023, doi:10.3390/molecules28227611_

Round 1

Reviewer 1 Report

Comments and Suggestions for Authors

The work of Xi and coauthors investigates the effect of Galactooligosaccharides (GOS) on intestinal microbiota and barrier pathway in antibiotic treated mouse model.

The manuscript provides useful data for the potential application of GOS enriched infant formula.

In order to improve the manuscript my recommendations are following:

Within the section 2.1. the intestinal microbiota mainly consisted of Firmicutes (Gram positive), Bacteroides (Gram negative), Proteobacteria (Gram negative), Actinobacteria (Gram positive).

Do you have data on changes of Bacteroides level at week 8? (line 87)

Similarly, do you have data about percentage changes in the level of Actinobacteria, besides Bacteroides at week 8? (line 89)

How do you explain that only one group go Gram negative bacteria decreased?

Please add to the legend of Figure 2 and Figure 3 explanation of abbreviations C, PC, PL, PM, MC, MM.

Section 2.3. (line 194): How do you support the statement that PC was accompanied by neutrophil infiltrations?

Line 240: Please add Figure 6 to the text

Recommendation: to modify to organization of the manuscript and move discussions on data from the results section to discussion section.

Reviewer 2 Report

Comments and Suggestions for Authors

The study by Xi M et al. aims to study the effects of GOS on intestinal microbiota and barrier function in mouse models treated with antibiotics using rRNA sequencing, gas chromatography and immunohistochemistry. The authors found a substantial beneficial effect of GOS at the level of molecular and histological alteration and microbiota composition.

The topic addressed by the authors is challenging and clinically relevant, the study design adequate and the method rather rigorous. However, the manuscript is difficult to read, the data are inadequately expressed in their presentation. 

A table describing the different experimental groups should be prepared by the authors to facilitate the experimental plan and description of results and understanding of figures and graphs. Excessive use of acronyms should be avoided.

Histology photomicrographies and immunohistochemistry are not informative in their current form, high-power fields are lacking, and the images do not really support the authors' findings.

Enlarged images with detailed description and indicators should be prepared by the authors.

The junctional complex should be studied more thoroughly, and other proteins tested immunohistochemically or molecularly such as claudins.

Lots of typos to fix.

Comments on the Quality of English Language

It needs to be improved.

Round 2

Reviewer 2 Report

Comments and Suggestions for Authors

Arrows in the figure are not currently described. Figures continue to be not informative as they could be if high power fields are be added. Please further intervene. 

The other issues raised have been properly addressed.

Comments on the Quality of English Language

Good
